# Rapid Classification of *Clostridioides difficile* Strains Using MALDI-TOF MS Peak-Based Assay in Comparison with PCR-Ribotyping

**DOI:** 10.3390/microorganisms9030661

**Published:** 2021-03-23

**Authors:** Adriana Calderaro, Mirko Buttrini, Monica Martinelli, Benedetta Farina, Tiziano Moro, Sara Montecchini, Maria Cristina Arcangeletti, Carlo Chezzi, Flora De Conto

**Affiliations:** 1Department of Medicine and Surgery, University of Parma, Viale A. Gramsci 14, 43126 Parma, Italy; mirko.buttrini@unipr.it (M.B.); benedetta.farina@studenti.unipr.it (B.F.); tiziano.moro@unipr.it (T.M.); mariacristina.arcangeletti@unipr.it (M.C.A.); carlo.chezzi@unipr.it (C.C.); flora.deconto@unipr.it (F.D.C.); 2Unit of Clinical Microbiology, University Hospital of Parma, Viale A. Gramsci 14, 43126 Parma, Italy; mmartinelli@ao.pr.it; 3Unit of Clinical Virology, University Hospital of Parma, Viale A. Gramsci 14, 43126 Parma, Italy; smontecchini@ao.pr.it

**Keywords:** *Clostridioides difficile*, MALDI-TOF MS, PCR-ribotyping

## Abstract

Typing methods are needed for epidemiological tracking of new emerging and hypervirulent strains because of the growing incidence, severity and mortality of *Clostridioides difficile* infections (CDI). The aim of this study was the evaluation of a typing Matrix-Assisted Desorption/Ionization-Time of Flight Mass Spectrometry (MALDI-TOF MS (T-MALDI)) method for the rapid classification of the circulating *C. difficile* strains in comparison with polymerase chain reaction (PCR)-ribotyping results. Among 95 *C. difficile* strains, 10 ribotypes (PR1–PR10) were identified by PCR-ribotyping. In particular, 93.7% of the isolates (89/95) were grouped in five ribotypes (PR1–PR5). For T-MALDI, two classifying algorithm models (CAM) were tested: the first CAM involved all 10 ribotypes whereas the second one only the PR1–PR5 ribotypes. Better performance was obtained using the second CAM: recognition capability of 100%, cross-validation of 96.6% and agreement of 98.4% (60 correctly typed strains, limited to PR1–PR5 classification, out of 61 examined strains) with PCR-ribotyping results. T-MALDI seems to represent an alternative to PCR-ribotyping in terms of reproducibility, set up time and costs, as well as a useful tool in epidemiological investigation for the detection of *C. difficile* clusters (either among CAM included ribotypes or out-of-CAM ribotypes) involved in outbreaks.

## 1. Introduction

*Clostridioides difficile* (previously named *Clostridium difficile*) is a Gram-positive, anaerobic, toxigenic, spore-forming bacterium, globally recognized as the pseudomembranous colitis aetiological agent and main causative pathogen of antibiotic-associated nosocomial infections [1]. During the last two decades, the epidemiological situation related to *C. difficile* infections (CDI) has been changing [2,3]. Clinical severity of CDI associated diseases has been increasing, and the related epidemiological scenario is evolving towards ever greater rates of incidence and mortality [4,5,6,7]. The growing severity of these infections is due to the emergence of new hypervirulent strains [8,9,10,11,12]. In particular, since 2003, the BI/NAP1/027 *C. difficile* strain became the most epidemiologically relevant strain in Europe, Northern and Central America, Asia, and Australia [13], most recently followed by the BK/NAP7/078 *C. difficile* strain [14]. The BI/NAP1/027 strain stands out for its greater pathogenicity: it is globally recognized as the main cause of nosocomial outbreak rising incidence and mortality in Europe and Northern America [15,16,17,18]. Given the epidemiological change currently occurring, molecular typing methods are needed for global epidemiological monitoring of CDI, and for identification and tracking of new emergent and hypervirulent strains, in order to enhance the global surveillance system of this phenomenon [19]. Different techniques are conventionally employed for *C. difficile* typing: restriction endonuclease analysis (REA), pulsed-field gel electrophoresis (PFGE), capillary or conventional agarose gel-based polymerase chain reaction (PCR) ribotyping (PCR-ribotyping), MultiLocus Variable-number tandem repeat analysis (MLVA), and MultiLocus sequence typing (MLST), as well as whole-genome sequencing (WGS) [20]. PCR-ribotyping and PFGE are the most used genotyping methods adopted in Europe and Northern America, respectively. PCR-ribotyping characterizes different *C. difficile* strains by the amplification of the Intergenic Spacer Region (ISR), located between 16S and 23S ribosomal genes, which has intraspecific high variability in terms of both length and nucleotide sequence; therefore, its variations identify different ribotypes [20,21]. PFGE is based on the catalytic activity of the *SmaI* restriction enzyme, which cuts *C. difficile* genomic DNA at specific restriction sites: the long DNA fragments obtained are separated by size, with an agarose gel electrophoresis on an electromagnetic field [20]. PCR-ribotyping proved to be sensitive and reliable for the identification of the epidemic strains; however, it shows a low discriminatory capability in differentiating strains having ISR of equal length, but different nucleotide sequences [22]. On the other hand, although PFGE has better discriminatory power than PCR-ribotyping, it is not able to thoroughly separate large DNA molecules, so errors can easily occur in the interpretation of results, especially when there are subtle differences between the control strain and the test strain [23,24]. Moreover, these traditional genotyping techniques are not advantageous in terms of costs and time of processing [23]. Growing CDI incidence, severity, and mortality require a reduction of costs and time-to-result in epidemiological tracking. PCR-ribotyping and PFGE cannot serve this purpose and, for this reason, new molecular typing methods are required. 

During the last decade, Matrix-Assisted Desorption/Ionization-Time of Flight Mass Spectrometry (MALDI-TOF MS) is catching on as a valid support in the workflow for laboratory diagnosis in clinical microbiology, especially for bacteriological identification [21] and virology [25,26]. More recently, thanks to its rapidity, accuracy and moderate price, MALDI-TOF MS has been employed as an alternative in the detection of antibiotic susceptibility/resistance biomarkers [27,28], in the identification of aminoacidic sequences and chemical structure of protein terminal groups [29], and as an emerging method in microbial typing [29,30]. 

To date, studies utilizing MALDI-TOF MS for typing of *C. difficile* strains have been performed using a low molecular weight (LMW) mass range (2–20 kDa) and focused only on few specific ribotypes, mainly involving the more virulent ones [21,31]. Moreover, two different studies described a MALDI-TOF MS method based on the high molecular weight (HMW) protein profile (mass range 30–50 kDa), including a larger number of ribotypes [32,33]. 

The aim of this study was, for the first time in our knowledge, to evaluate the possibility to differentiate and classify by MALDI-TOF MS the different toxigenic *C. difficile* strains circulating in our area during the clinical practice using a statistical classifying algorithm model (CAM) in comparison with the PCR-ribotyping results.

## 2. Materials and Methods

### 2.1. Samples 

A total of 1486 faecal samples sent to the Unit of Clinical Microbiology of the University Hospital of Parma (Italy) for diagnostic purpose belonging to 1208 patients with suspicion of CDI, as reported in the medical order, during a 10-month period (from November 2018 to March 2019 and from October 2019 to February 2020) were included.

Laboratory diagnosis was performed upon medical request and a clinical report was produced. Anonymization of patients was done before data analysis and medical information were protected.

### 2.2. Study Design

All stool samples were prospectively analysed for the detection of toxigenic *C. difficile* by a two-step diagnostic algorithm, as previously reported [34]. Briefly, the first step involved a molecular qualitative assay (Illumigene^TM^
*C. difficile*, Meridian Bioscience, Cincinnati, OH, USA), based on a loop-mediated isothermal DNA amplification (LAMP) technology, able to detect toxigenic *C. difficile* by amplifying a 204 bp nucleotide sequence inside *C. difficile*’s PaLoc that is located at the 5’ region of the tcdA gene and conserved in all known toxinotypes. The assay was performed according to the manufacturer’s instructions, as previously described [34].

The second step, performed only on toxigenic *C. difficile* DNA positive samples, involved the simultaneous detection of the glutamate dehydrogenase enzyme (GDH) and of the toxins A/B by an immunochromatographic assay (C. DIFF QUICK CHECK COMPLETE TechLab, Blacksburg, VA, USA), performed according to the manufacturer’s instructions as previously described [34]. In parallel, the samples were submitted to *C. difficile* isolation by culture (CC). Briefly, an aliquot of faecal sample was added to an enrichment medium (Cooked meat broth, Kima, Padova, Italy), incubated at 37 °C in anaerobic conditions (95% N, 5% CO_2_) for 72 h and then heat-shocked (100 °C for 3 min) before plating onto a specific selective medium (cycloserine-cefoxitin-fructose agar—CCFA, Kima). After incubation at 37 °C in anaerobic conditions for at least 48 h, the species identification of putative *C. difficile* colonies was performed by a MALDI-TOF mass spectrometer (Bruker, Bremen, Germany) [34]. A 2.5 McFarland suspension in 1 mL of sterile double-distilled water of each *C. difficile* isolate from CCFA culture was used for typing by PCR-ribotyping and MALDI-TOF MS. 

### 2.3. Typing of C. difficile Isolates 

#### 2.3.1. PCR-Ribotyping

For PCR-ribotyping, an aliquot of 400 μL of the bacterial suspension was treated by heat shock at 100 °C for 10 min [34] and the supernatant containing *C. difficile* DNA was stored at 4 °C until the amplification of the ISR by using two specific primers (Eurogentec, Seraing, Belgium), previously described [35]: RtFR1 (5’- GTG CGG CTG GAT CAC CTC CT-3’) complementary to the 3’ terminal region of the 16S ribosomal gene and RtFR2 (5’-CCC TGC ACC CTT AAT AAC TTG ACC-3’) complementary to the 5’ terminal region of the 23S ribosomal gene. Amplification reaction was performed according to Bidet et al. [35] with some modifications: an aliquot of 15 μL of *C. difficile* DNA was added to a 35-μL reaction mixture containing PCR Buffer 1X (5 μL) (Roche, Monza, Italy), 1.5 mM MgCl_2_ (Roche), 10 pmol of each primer, 200 μM of dNTPs (Roche), and 1.5 U TaqDNA polymerase (Roche). The amplification was carried out in a GeneAmp PCR System 9700 thermalcycler (Applied Biosystems, Foster City, CA, USA), according to the following protocol: one cycle of 6 min at 95 °C; 35 cycles of 1 min at 94 °C, 1 min at 57 °C, and 2 min at 72 °C; and a final extension cycle of 7 min at 72 °C. Amplification products (10 μL added to 2 μL of bromophenol blue, Invitrogen, Paisley, UK) were separated by electrophoresis through a 3% agarose gel in Tris-acetate-EDTA buffer for 5 h at 85 V and revealed on a UV table after GelRed^®^ staining. Gel images were acquired digitally. Strains differing for at least two genetic bands were assigned to different groups, arbitrarily numbered in a progressive way (i.e., PR1, PR2,…).

#### 2.3.2. MALDI-TOF MS for Typing (T-MALDI)

For MALDI-TOF MS protein extraction, an aliquot of 300 μL of the same bacterial suspension used for PCR-ribotyping was added to 900 μL of absolute ethanol, homogenized and then centrifuged at 14.000× *g* for 2 min. Fifteen μL of 70% formic acid and 15 μL of acetonitrile were added to the pellet, previously dried for at least 5 min under a laminar flow cabinet at room temperature, then vortexed (20 s) and centrifuged (14.000× *g* for 2 min). One μL of the supernatant containing the protein extract was transferred on a MALDI-TOF target plate (10 replicates for each strain), dried at room temperature and then overlaid with 1 μL of α-Cyano-4-hydroxycinnamic acid Matrix (HCCA), solubilized onto Organic Solvent (OS-TA30, with ratio 30:70 of acetonitrile/trifluoroacetic acid 0.01%). Finally, the 10-dried spots of each sample were analysed by an Autoflex Speed mass spectrometer (Bruker) using the MBT_Standard method (positive linear mode, with 60 Hz laser frequency, ion source voltage 20 kV and mass molecular range 2–20 kDa). Each spot was acquired in manual mode, in different points of the well with a laser intensity ranging from 30 to 40%, with an overall 1400 laser-shot, by 200 shot steps. MALDI-TOF MS calibration was performed for each run with “Bruker Bacterial Test Standard (BTS)” (Bruker), according to the manufacturer’s instructions. The acquired spectra were analysed by FlexAnalysis software (version 3.1, Bruker), normalized by “Smoothing” and “Baseline” functions. The spectra with <10^4^ intensity arbitrary units were removed. Spectra analysed by FlexAnalysis were then identified by Biotyper software (version 3.1.66, Bruker) in order to verify their validity, and to identify the bacterial species. Only the 10-replicates sets identified as *C. difficile* with a >2-score value were used for further analysis.

For MALDI-TOF MS typing (T-MALDI), the spectra of the strains acquired by MALDI-TOF MS were retrospectively analysed, after PCR-ribotyping results, by ClinProTools software (version 3.0, Bruker), in order to detect markers able to discriminate the different *C. difficile* types. The analysis was focused on the molecular mass range 2–20 kDa, with a 7.5 signal-to-noise ratio, and a 0.75 noise threshold. All spectra were automatically re-calibrated, with “Shift Maximum Peak” set up at 1000 ppm, to reduce the mass shifts that could arise during multiple acquisitions. An average spectrum based on the replicates of each isolate (single average spectrum), as well as an average spectrum based on all replicates of all analysed isolates (cumulative average spectrum), were created. The average spectra analysis provided a list of peaks, potentially discriminating the different *C. difficile* types, in combination with a *p*-value. 

To create a classifying algorithm model (CAM), the Genetic Algorithm (GA), the Quick Classifier (QC), and the Supervised Neural Network (SNN) algorithm-based models were compared using a training set of strains on the basis of the PCR-ribotyping results. Each algorithm automatically selected a restricted pattern of peaks among those previously found, able to classify the strains. Each CAM was characterized by recognition capability (RC) and cross-validation (CV) values, parameters of the accuracy of the model. The algorithm with the highest RC and CV scores was chosen as CAM for this study. 

#### 2.3.3. Statistical Analysis

The *p*-value was calculated by comparing each single average spectrum with the cumulative average spectrum, based on both parametric (Analysis of Variance—ANOVA) and non-parametric (Kruskal-Wallis—KW) statistical tests. In this study, a *p*-value < 0.001 was considered significant.

## 3. Results

During the study period, toxigenic *C. difficile* DNA was revealed in 158 patients (13.1%, 158/1208). Among the 158 *C. difficile* DNA positive patients, GDH was revealed in 151 patients (95.6%), in 79 cases in combination also with toxins A/B (50%, 79/158). Ninety-five *C. difficile* isolates (60.1%, 95/158) were obtained by CC and typed by PCR-ribotyping. PCR amplification pattern analysis revealed 10 different ribotypes, arbitrarily named PR1–PR10 (Figure 1 and Figure 2). 

In particular, 93.7% of the isolates (89/95) were grouped in 5 ribotypes (PR1–PR5) and PR1 (42.1%, 40/95) was the most represented one.

For T-MALDI, spectra acquired were imported, according to the PCR-ribotyping results, to the dedicated program in order to obtain a pattern of protein peaks able to discriminate the PR1–PR10 ribotypes. Thirty-nine out of the 95 strains were arbitrarily selected as training set: eight strains for each PR1–PR4 ribotype, two strains for PR5 ribotype and one strain for each PR6-PR10 ribotype. When the different algorithms (GA, QC and SNN) were applied on 122 potential discriminant peaks initially identified, the GA-based CAM showed the best performances in terms of RC (98.75% vs. 48.9% and 57.5% for QC and SNN, respectively) and CV (91.6% vs. 47.1% and 50.2% for QC and SNN, respectively), selecting 17 peaks with a *p*-value < 0.0001 for both ANOVA and KW statistical tests (Figure 3, Appendix A).

To verify the reliability and the accuracy of this CAM, internal and external validations were performed. In particular, the 39 strains arbitrarily included in the training set were used as internal controls, while the remaining 56 strains (32 PR1, 12 PR2, 4 PR3, 6 PR4, 1 PR5, and 1 PR6) were used as external controls. All the 39 internal control strains and 25 (16 PR1, 3 PR2, 2 PR3, 3 PR4 and 1 PR5) out of the 56 (44.6% 25/56) external control strains were correctly classified in agreement with the PCR-ribotyping results. Among the remaining 31 external control strains (55.4%), 18 (32.1%, 18/56; 9 PR1, 5 PR2, 1 PR3, and 3 PR4) were not classified in any ribotype considered and 13 (23.2% 13/56; 7 PR1, 4 PR2, 1 PR3 and 1 PR6) were classified in a ribotype different from that assigned by PCR-ribotyping. In particular, for these 13 latter strains, the 7 PR1 were classified as PR3 in 4 cases, as PR2 in 2 cases and as PR6 in 1 case; the 4 PR2 strains were classified as PR3 in all 4 cases; both PR3 and PR6, with one strain each, were classified as PR2. 

A second GA-based CAM was developed focusing on the five most frequent ribotypes (PR1–PR5). Starting from a training set of 34 strains (8 for each PR1–PR4 and 2 for PR5) out of the 39 used to create the first model, 60 potential discriminating peaks were initially identified. The overall RC (100%) and CV (96.6%) rates were higher than those obtained for the first CAM (Table 1) and a total of 17 peaks, 13 of which differ from those of the first one, were selected as the most discriminants (*p*-value of each peak <0.0001, for both ANOVA and KW statistical tests) (Figure 4 and Appendix A).

To verify the reliability and the accuracy of this second CAM, internal and external validations were performed using the 34 training set strains as internal controls and the remaining 61 (32 PR1, 12 PR2, 4 PR3, 6 PR4, 1 PR5, 2 PR6, 1 PR7, 1 PR8, 1 PR9 and 1 PR10) as external controls. All the 34 internal control strains were correctly classified in agreement with PCR-ribotyping results. With regard to the external validation, T-MALDI results for the PR1–PR5 ribotypes were in agreement with those of PCR-ribotyping in all cases except one PCR-ribotype PR4 (60/61, 98.4%), which was not classified among the five ribotypes considered (Table 2).

## 4. Discussion

Antibiotic misuse and emergence of new hypervirulent *C. difficile* strains, mainly in nosocomial environments, led to a global increase in CDI incidence rate [5,16,36] with a mortality rate ranging from 3% to 30% [18]. In this study, 158 CDI cases were detected with an overall observed *C. difficile* prevalence of 13.1%. This prevalence rate was higher than that pointed out by the European Centre for Disease Prevention and Control [36], assessed to be 9.9% for Italy; however, this value could not be representative since only two hospitals were included with a total of 12 CDI detected cases. 

In our study, PCR-ribotyping performed on 95 *C. difficile* strains revealed 10 different ribotypes (arbitrarily named PR1–PR10), mainly grouped in five ribotypes (PR1–PR5), accounting for 93.7% (89/95). This distribution could suggest that there are no significant variations in the spread of different ribotypes in our hospital, even if for a limited study period. 

Although no internationally recognized ribotype was assigned to each ribotype identified during this study, the most frequently detected PR1 and PR2 ribotypes (60 out of 95) showed a PCR-ribotyping pattern similar to the well-characterized RT018 and RT126 ribotypes, respectively, already found as the most prevalent in a previous study performed in our area (Unpublished data). Moreover, the PR1/018-like was the most frequent ribotype, in agreement with other Italian epidemiological studies, in which it was also recognized as the most frequent cause of nosocomial CDI in the elderly [37,38,39].

In order to evaluate the possibility to differentiate and classify by MALDI-TOF MS different *C. difficile* strains, two CAMs were created based on the GA algorithm. The first attempt was performed using a selection of 39 strains representative of all 10 ribotypes, including five strains belonging to the more rarely detected ribotypes (PR6–PR10). Despite the high values of RC and CV (98.75% and 91.6%, respectively), only 44.6% (25/56) of the external control strains were correctly classified. The second GA-based CAM was generated using the same strains employed for the first one, except the five strains belonging to the more rare ribotypes, showing RC and CV values of 100% and 96.6%, respectively. When the 61 external control strains were analysed, 98.4% (60 out of 61) were correctly classified, in agreement with PCR-ribotyping with regard to the ribotypes used for the CAM creation.

The better performance of the second CAM was likely due to the different selection of ribotypes in the training set, which included only the five most frequent ribotypes, excluding those used in the first CAM for which only one strain was detected. Generally, the classification models work optimally with a high number of strains for each ribotype. In fact, the fewer the number of strains used to create the ribotype average spectrum, the lower the reliability of the discriminant peaks obtained for that specific ribotype. Furthermore, when single-strain ribotypes are involved in the development of a model, the CAM could include among the discriminant peaks a protein that would not be discriminant for a specific ribotype but rather only for a single strain. The potential genetic variability of *C. difficile* strains within the same ribotype and the emergence of new circulating ribotypes could give reason for the failure of T-MALDI in the classification of the ribotypes not included in the CAM, information about the presence of *C. difficile* strain clusters would be available in any case. However, for the correct classification, a model update would be necessary.

Furthermore, a standardized inter-laboratory CAM could be affected by the lack of a T-MALDI international reference database for *C. difficile* typing and the missing association between reference ribotypes and specific marker peaks could hamper the correct classification of the circulating strains with particular reference to the hypervirulent ones. In addition, global inter-laboratory data sharing is problematic due to the local different experimental protocols (i.e. growth medium, protein extraction, and concentration of acids added to the matrix) and instrument hardware and software that could influence the detection of the CAM discriminating peaks, as already reported [29].

Besides previous studies focusing on MALDI-TOF typing methods based on LMW (2–20 kDa) for the characterization of the most virulent ribotypes, such as 027 and 078, further HMW (30–50 kDa) and proteotyping studies [21,32,33,40] investigated the possibility to identify protein biomarkers uniquely detecting currently recognized ribotypes. Although the ribotypes differentiation can be obtained in either LMW or HMW range, when considering a high number of ribotypes, the best discriminating power in comparison to PCR-ribotyping was obtained by the combined analysis with both molecular ranges (2–20 kDa and 30–50 kDa) [33]. As a matter of fact, in our study the LMW-range-based CAM correctly differentiated the five predominant ribotypes without changes in the identification acquisition protocol; however, a CAM covering a larger number of ribotypes could not be able to differentiate two or more ribotypes with the same protein profile in LMW. In this case, the discriminating power of a CAM could be improved by involving spectra acquired in the HMW range, with a dedicated acquisition protocol. 

Moreover, the discriminating power of a CAM is highly related to the quality of the ribotype average spectrum, in terms of presence/absence, intensity and numbers of specific peaks revealed. In particular, variations in matrices, the concentration of acids added to the matrix and spectra acquisition during different growth phases could increase the number of peaks in order to differentiate a larger number of ribotypes. Therefore, further analysis should assess the impact of analytical and pre-analytical variables on the pattern of peaks detected in order to find an improvement of the spectrum quality, even if for this purpose a specific protocol should be developed before the experimental approach.

Despite these few limits, in our hand, MALDI-TOF MS technology for *C. difficile* typing proved to be useful in supporting the epidemiological investigation performed by PCR-ribotyping in this study. In the light of our results, the CAM with the best discriminant power, even if created on few ribotypes (those with a greater number of strains per ribotype), did not affect the epidemiological tracking, failing only on a limited number of cases involving ribotypes more rarely detected.

Thus, T-MALDI for *C. difficile* classification could be a valid alternative to PCR-ribotyping. In particular, the protein spectra acquisition for MALDI-TOF MS typing in comparison with the PCR-ribotyping turned out to be easier (only a few manual steps and minimum hands-on time for spectra acquisition vs. several labour-intensive steps for nucleic acid amplification, DNA fragments separation and PCR amplification pattern analysis), faster (30 min vs. at least 12 h) and cheaper (€1.5 vs. €15, for reagents and disposable materials per each strain). Moreover, T-MALDI is suitable also for a single-strain analysis, allowing real-time monitoring of *C. difficile* circulating strains, whereas PCR-ribotyping is optimized for the analysis of many samples in a batch.

## 5. Conclusions

In conclusion, although a validated CAM could not perform well in the classification of *C. difficile* strains during longer study periods, it can still represent a useful epidemiological tool for the classification of the most frequently circulating ribotypes, such as those detected during a short period, and for the detection of epidemiologically-linked *C. difficile* clusters involved in outbreaks, even if related to ribotypes not included in CAM.

## Figures and Tables

**Figure 1 microorganisms-09-00661-f001:**
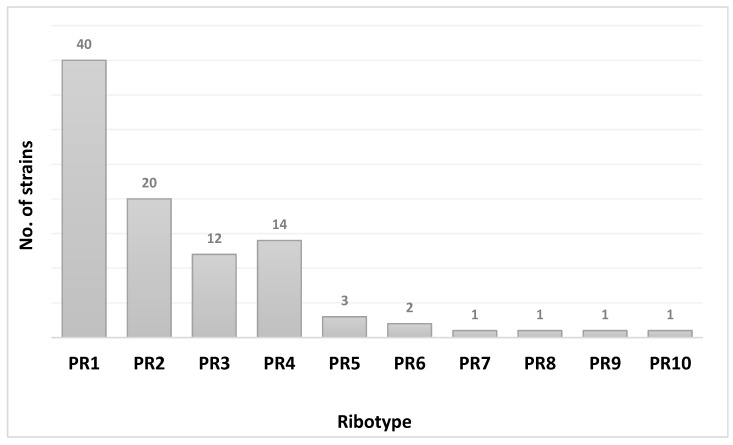
Different *Clostridioides*
*difficile* ribotypes found in the study period.

**Figure 2 microorganisms-09-00661-f002:**
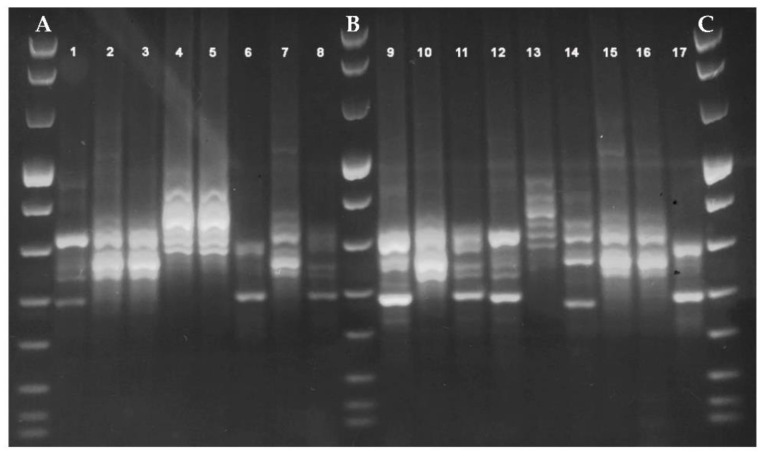
Example of different polymerase chain reaction (PCR)-ribotyping patterns. Lines 2, 3, 7, 10, 15, 16: Ribotype PR1; Lines 1, 8, 11, 12: Ribotype PR2; Lines 6, 17: Ribotype PR3; Lines 4, 5, 13: Ribotype PR4; Line 9: Ribotype PR6; Line 14: Ribotype PR7; Lines A, B, C: 100-bp ladder DNA molecular weight VIII (Roche).

**Figure 3 microorganisms-09-00661-f003:**
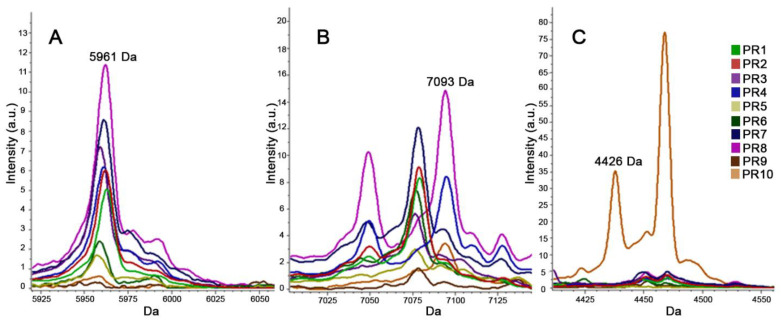
Examples of discriminating peaks able to differentiate the 10 different ribotypes found. (**A**): peak of 5961 Dalton present in all ribotypes except in PR9 and PR10. (**B**): peak of 7093 Dalton present in ribotypes PR4, PR7, PR8, and PR10. (**C**): peak of 4426 Dalton present only in ribotype PR10. The spectra of the ribotypes PR1, PR2, PR3, PR4, PR5, PR6, PR7, PR8, PR9, and PR10 are shown in green, red, purple, blue, yellow, dark green, dark blue, pink, brown, and orange, respectively.

**Figure 4 microorganisms-09-00661-f004:**
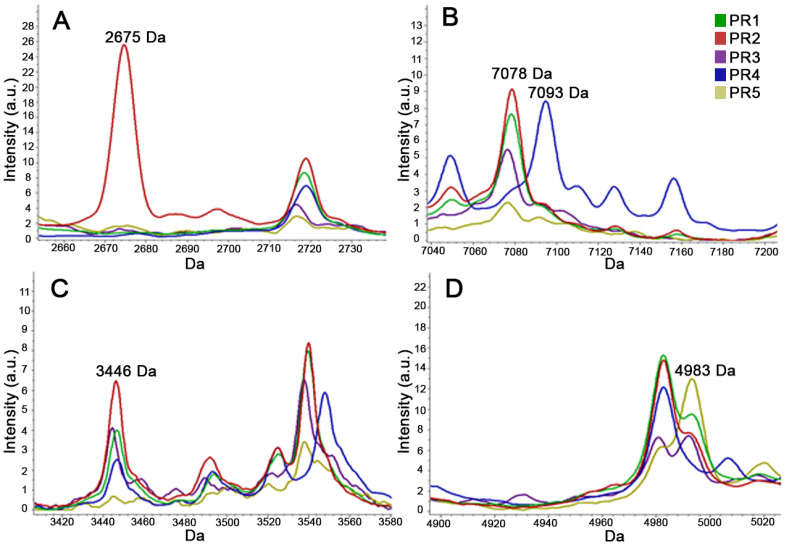
Examples of discriminating peaks able to differentiate the PR1–PR5 ribotypes. (**A**): peak of 2675 Dalton present only in ribotype PR2; (**B**): peaks of 7078 Dalton present in all ribotypes except PR4 and of 7093 Dalton present only in ribotype PR4; (**C**): peak of 3446 Dalton present in ribotypes PR1, PR2, PR3, and PR4; (**D**): peak of 4983 Dalton present in ribotypes PR1, PR3, and PR5. The spectra of the ribotypes PR1, PR2, PR3, PR4, and PR5 are shown in green, red, purple, blue, and yellow, respectively.

**Table 1 microorganisms-09-00661-t001:** Recognition capability (RC) and cross-validation (CV) values of the classifying algorithm model (CAM) able to classify 10 ribotypes (PR1–PR10) and the CAM able to classify the five most frequent ribotypes (PR1–PR5).

Ribotype	PR1–PR10 CAM		PR1–PR5 CAM	
	RC (%)	CV (%)	RC (%)	CV (%)
**PR1**	100	96.3	100	99.5
**PR2**	98.7	95.1	100	92.5
**PR3**	98.7	95.6	100	100
**PR4**	100	91.5	100	95.5
**PR5**	100	95.6	100	95.8
**PR6**	100	78.3	NA	NA
**PR7**	90	72.7	NA	NA
**PR8**	100	95.8	NA	NA
**PR9**	100	94.7	NA	NA
**PR10**	100	100	NA	NA
**Overall**	98.7	91.6	100	96.6

NA: not applicable.

**Table 2 microorganisms-09-00661-t002:** Matrix Assisted Desorption/Ionization-Time of Flight Mass Spectrometry (MALDI-TOF MS) typing (T-MALDI) results performed on 61 *C. difficile* strains based on the second classifying algorithm model.

PCR-Ribotyping	T-MALDI					
	Ribotype PR1	Ribotype PR2	Ribotype PR3	Ribotype PR4	Ribotype PR5	No Classification
Ribotype PR1	32					
Ribotype PR2		12				
Ribotype PR3			4			
Ribotype PR4				5		1
Ribotype PR5					1	
Ribotypes PR6-PR10						6

## Data Availability

The data presented in this study are available in the manuscript and in the Appendix A.

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
