# Peer review of "Rapid Classification of Clostridioides difficile Strains Using MALDI-TOF MS Peak-Based Assay in Comparison with PCR-Ribotyping"

_microorganisms, 2021, doi:10.3390/microorganisms9030661_

Round 1
Reviewer 1 Report
To explore the use of MALDI-TOF as an alternative to PCR ribotyping for typing C. difficile, Calderaro et al screen fecal samples from suspected CDI patients for toxigenic C. difficile. The resulting isolates are internally PCR ribotyped and subjected to MALDI-TOF analysis. To generate a predictive algorithm for categorizing unknown samples, two different models are tested. The model containing fewer ribotypes with more representative strains per ribotype provided better accuracy. The manuscript is well written and clear.
The limitations of this study are two-fold: 1) using an arbitrary PCR ribotype numbering system without verifying these data to an international PCR ribotyping standard or database. The absence of this data results in missing out on potentially interesting data about the epidemiology or circulating ribotypes at the time of collection (RT018 and RT0126 (?) mentioned in Discussion with data not shown). Accurate PCR ribotyping information, at least of the five most prevalent ribotypes identified, would strengthen the impact and the interest in the study as would provide local surveillance information.
And 2) the MALDI-TOF analysis covers only lower molecular weight peaks (2-20 kDa). Less discriminant typing with lower molecular weight ranges has already been observed in previous studies, which found more specific and higher typing resolution at higher molecular weight ranges (30-50 kDa). This is addressed somewhat in the discussion, but perhaps methods combining both lower and higher molecular weight ranges (and deriving a CAM from these) will provide better resolution and the best accuracy so far for typing unknown C. difficile strains by MALDI-TOF. Discussing further improvements to this model and emphasizing what future studies can be performed to aid in the development of a CAM covering a larger number of ribotypes would clarify what impact this study has in the implementation of this technique in the C. difficile field.
Including a more detailed background/discussion on C. difficile MALDI-TOF typing in the introduction and/or Discussion would place this study in the context of work already done in the field.
Minor comments
Line 186: Qualifying for presence of both Toxin A and Toxin B or only one of two toxin present? If using C. Diff Quick Check Complete here, clarify Toxin A and/or Toxin B.
Line 82: By screening for the presence of toxin, this current study also focuses on only virulent C. difficile isolates. Also, other studies, such as Rizzardi and Akerlund and Ortega et al, correlated dozens of ribotypes with (mostly) resolved MALDI-TOF spectra (also mentioned in lines 299-301).
Author Response
Reviewer 1
To explore the use of MALDI-TOF as an alternative to PCR ribotyping for typing C. difficile, Calderaro et al screen fecal samples from suspected CDI patients for toxigenic C. difficile. The resulting isolates are internally PCR ribotyped and subjected to MALDI-TOF analysis. To generate a predictive algorithm for categorizing unknown samples, two different models are tested. The model containing fewer ribotypes with more representative strains per ribotype provided better accuracy. The manuscript is well written and clear.
We thank the Reviewer for the appreciation of our work and for her/his effort in suggesting to improve the manuscript.
The limitations of this study are two-fold: 1) using an arbitrary PCR ribotype numbering system without verifying these data to an international PCR ribotyping standard or database. The absence of this data results in missing out on potentially interesting data about the epidemiology or circulating ribotypes at the time of collection (RT018 and RT0126 (?) mentioned in Discussion with data not shown). Accurate PCR ribotyping information, at least of the five most prevalent ribotypes identified, would strengthen the impact and the interest in the study as would provide local surveillance information.
The only two internationally assigned PCR-ribotyping patterns available from our collection are RT018 and RT126, already found as the most frequently revealed and characterized in a previous study performed in our area. The comparison of these two well-known PCR-ribotyping patterns with those obtained in this study revealed that the arbitrarily-assigned PR1 and PR2 ribotypes (60 out of 95) corresponded to RT018 and RT126, respectively. This point was clarified in the discussion section (Page 8, lines 269-273).
And 2) the MALDI-TOF analysis covers only lower molecular weight peaks (2-20 kDa). Less discriminant typing with lower molecular weight ranges has already been observed in previous studies, which found more specific and higher typing resolution at higher molecular weight ranges (30-50 kDa). This is addressed somewhat in the discussion, but perhaps methods combining both lower and higher molecular weight ranges (and deriving a CAM from these) will provide better resolution and the best accuracy so far for typing unknown C. difficile strains by MALDI-TOF. Discussing further improvements to this model and emphasizing what future studies can be performed to aid in the development of a CAM covering a larger number of ribotypes would clarify what impact this study has in the implementation of this technique in the C. difficile field.
According to the reviewer’s suggestion, the potential discriminating capabilities of a CAM both in lower and higher molecular weight were discussed in the discussion section (Page 9, lines 312-332).
Including a more detailed background/discussion on C. difficile MALDI-TOF typing in the introduction and/or Discussion would place this study in the context of work already done in the field.
As suggested, details on C. difficile MALDI-TOF typing were added both in background and discussion sections (Page 2, lines 81-86 and page 9 lines 312-324).
Minor comments
Line 186: Qualifying for presence of both Toxin A and Toxin B or only one of two toxin present? If using C. Diff Quick Check Complete here, clarify Toxin A and/or Toxin B.
The sentence was modified accordingly.
Line 82: By screening for the presence of toxin, this current study also focuses on only virulent C. difficile isolates. Also, other studies, such as Rizzardi and Akerlund and Ortega et al, correlated dozens of ribotypes with (mostly) resolved MALDI-TOF spectra (also mentioned in lines 299-301).
The sentence was modified accordingly.

Reviewer 2 Report
This work by Calderaro describes a practical method for typing strains of C. difficile which is comparable to ribotyping, but more convenient and inexpensive (for sites that already have MALDI-TOF MS). The authors were honest in demonstrating that their first attempt at characterizing strains according to the top 10 ribotypes failed validation; they postulate that there were too few samples in some of the ribotypes to generate a reliable correlate among the MALDI data. When they restricted the assignment to the top 5 ribotypes, with all others lumped together, they were able to correctly assign 60 of 61 isolates in their validation cohort. Interestingly, in the 2nd analysis, 13 of the 17 protein peaks assessed were different from the 1st analysis, pointing to the importance of the model selection, and potential pitfalls in cross- validation.
The work demonstrates that this typing method can be a robust means of identifying locally linked isolates of C difficile - my concern for this method to be used more broadly is whether the MALDI data generated is reliable across different sites in different regions of the globe - i.e. do all isolates of a particular ribotype have the same MALDI pattern? There is no a priori reason to expect that they would, as ribotype and MALDI measure 2 distinct parameters of diversity that are not necessarily perfectly aligned. But this work and others referenced within suggest it may be worth evaluating this approach. Did the authors have access to any reference strains (027, 078) to try to validate their MALDI data against that of others?
Minor comments for the authors to consider changing
Page 1, line 17 : cause ->because
Page 1, line 38 – is constantly ->has been
Page 2 line 62 –infectious strains contextualizing to epidemic events->epidemic strains
Page 2 line 66 –delete remarkably
Page 2, line 71 – answer properly at this ->serve
Page 2, line 76 – cheapness->moderate price
Page 8 line 275 –the 44.6%->delete “the”
Page 8, line 279- the 98.4% -> delete “the”
Page 8, line 282- The best performances.. were->The better performance…was
Page 8, line 287 – fewer->lesser
Page 9, line 290-Delete Although
Page 9, line 292 –reason of the failure->reason for the failure
Author Response
Reviewer 2
This work by Calderaro describes a practical method for typing strains of C. difficile which is comparable to ribotyping, but more convenient and inexpensive (for sites that already have MALDI-TOF MS). The authors were honest in demonstrating that their first attempt at characterizing strains according to the top 10 ribotypes failed validation; they postulate that there were too few samples in some of the ribotypes to generate a reliable correlate among the MALDI data. When they restricted the assignment to the top 5 ribotypes, with all others lumped together, they were able to correctly assign 60 of 61 isolates in their validation cohort. Interestingly, in the 2nd analysis, 13 of the 17 protein peaks assessed were different from the 1st analysis, pointing to the importance of the model selection, and potential pitfalls in cross- validation.
We thank the Reviewer for the appreciation of our work and for her/his effort in suggesting to improve the manuscript.
The work demonstrates that this typing method can be a robust means of identifying locally linked isolates of C difficile - my concern for this method to be used more broadly is whether the MALDI data generated is reliable across different sites in different regions of the globe - i.e. do all isolates of a particular ribotype have the same MALDI pattern? There is no a priori reason to expect that they would, as ribotype and MALDI measure 2 distinct parameters of diversity that are not necessarily perfectly aligned. But this work and others referenced within suggest it may be worth evaluating this approach. Did the authors have access to any reference strains (027, 078) to try to validate their MALDI data against that of others?
According to the reviewer’s suggestion, the inter-laboratory reproducibility was more deeply discussed (Page 9, lines 303-311).
Unfortunately, at the moment it is not allowed to access to viable cultures of RT 027 and 078 or other laboratory reference strains. For the strain RT027 only the PCR-Ribotyping pattern is available in our collection. Despite we are aware about the epidemiological relevance of these two strains, in this study our approach was based on the assessment of the capabilities of MALDI-TOF MS, using a statistical classifying algorithm model, to discriminate among ribotypes circulating and randomly collected in our area.
Minor comments for the authors to consider changing
Page 1, line 17 : cause ->because
The change was done.
Page 1, line 38 – is constantly ->has been
The change was done.
Page 2 line 62 –infectious strains contextualizing to epidemic events->epidemic strains
The sentence was modified accordingly.
Page 2 line 66 –delete remarkably
The change was done.
Page 2, line 71 – answer properly at this ->serve
The sentence was modified accordingly.
Page 2, line 76 – cheapness->moderate price
The change was done.
Page 8 line 275 –the 44.6%->delete “the”
The change was done.
Page 8, line 279- the 98.4% -> delete “the”
The change was done.
Page 8, line 282- The best performances.. were->The better performance…was
The sentence was modified accordingly.
Page 8, line 287 – fewer->lesser
The change was done.
Page 9, line 290-Delete Although
The change was done.
Page 9, line 292 –reason of the failure->reason for the failure
The change was done.

Round 2
Reviewer 1 Report
This manuscript is much improved with the additional discussion regarding limitations of the current study and how their CAM may be improved to easily, quickly and cost effectively identify ribotypes isolated from clinical samples. This discussion clearly places this study in the context of the field and the future of using MALDI-TOF for C. difficile classification.
Line 272: Please cite the previous study performed in your area that found RT018/RT126 as the most prevalent ribotypes.
Author Response
Reviewer 1
This manuscript is much improved with the additional discussion regarding limitations of the current study and how their CAM may be improved to easily, quickly and cost effectively identify ribotypes isolated from clinical samples. This discussion clearly places this study in the context of the field and the future of using MALDI-TOF for C. difficile classification.
We thank the Reviewer to have recognized and appreciated the improvement to our discussion, based on his/her observations.
Line 272: Please cite the previous study performed in your area that found RT018/RT126 as the most prevalent ribotypes.
The cited study referred to a retrospective epidemiological investigation on the results of C. difficile PCR-Ribotyping strains circulating in our area. In that study, the strains representative of the two most frequently detected Ribotypes were sent to Cardiff reference laboratory and confirmed as PCR-ribotype 126 and 018. Although a manuscript related to this study was submitted to an international journal, this remains still unpublished. For this reason, “Data not shown” was replaced with “Unpublished data”.